# Research and Public Interest in Mindfulness in the COVID-19 and Post-COVID-19 Era: A Bibliometric and Google Trends Analysis

**DOI:** 10.3390/ijerph20053807

**Published:** 2023-02-21

**Authors:** Chan-Young Kwon

**Affiliations:** Department of Oriental Neuropsychiatry, Dong-Eui University College of Korean Medicine, 52-57, Yangjeong-ro, Busanjin-gu, Busan 47227, Republic of Korea; beanalogue@deu.ac.kr

**Keywords:** mindfulness, COVID-19, bibliometric analysis, VOSviewer, Google trends

## Abstract

Public and research interest in mindfulness has been growing, and the Coronavirus disease 2019 (COVID-19) pandemic seems to have accelerated this growth. This study was conducted to investigate the public and research interest in mindfulness in the context of COVID-19. The term ‘Mindfulness’ was searched in Google Trends, and data were collected from December 2004 to November 2022. The relationship between the relative search volume (RSV) of ‘Mindfulness’ and that of related topics was analyzed, and ‘Top related topics and queries’ for the search term ‘Mindfulness’ were investigated. For bibliometric analysis, a search was conducted in the Web of Science database. Keyword co-occurrence analysis was conducted, and a two-dimensional keyword map was constructed using VOSviewer software. Overall, the RSV of ‘Mindfulness’ increased slightly. The RSVs of ‘Mindfulness’ and ‘Antidepressants’ showed an overall significant positive correlation (r = 0.485) but a statistically significant negative correlation during the COVID-19 era (−0.470). Articles on mindfulness in the context of COVID-19 were closely related to depression, anxiety, stress, and mental health. Four clusters of articles were identified, including ‘mindfulness’, ‘COVID-19’, ‘anxiety and depression’, and ‘mental health’. These findings may provide insights into potential areas of interest and identify ongoing trends in this field.

## 1. Introduction

Mindfulness, defined by Jon Kabat-Zinn as “paying attention in a particular way: on purpose, in the present moment, and non-judgmentally” [1], has been reported to be associated with a variety of psychological benefits [2]. Mindfulness training has also been reported to benefit individuals with physical illnesses such as chronic pain, common cold, psoriasis, irritable bowel syndrome, and diabetes [3]. According to an analysis of the 2012 National Health Interview Survey, 2.5% of American adults practice mindfulness meditation in their lifetime, which represents 5.7 million people [4]. Structured mindfulness training programs such as mindfulness-based stress reduction and mindfulness-based cognitive therapy (MBCT) have contributed to the popularization of mindfulness [5]. The popularity of mindfulness in modern society is related to increased stress and anxiety in modern society, as well as a growing public interest in well-being [6]. The popularity of mindfulness is extending not only to individuals, but also to businesses and organizations (e.g., corporate mindfulness) [7,8]. In addition, mindfulness training has recently become more popular in the general population, especially in the form of smartphone applications, in conjunction with information and communication technology (ICT) [9].

Coronavirus disease 2019 (COVID-19) has highlighted the importance of mental health worldwide since the end of 2019. According to a comprehensive systematic review of the prevalence of mental health problems during the COVID-19 pandemic, the overall pooled prevalence of depression, anxiety, distress, and insomnia reached 31%, 32%, 41%, and 38%, respectively, among all types of target populations (e.g., general population, medical staffs, students, patients with COVID-19, and patients with other clinical conditions) [10]. Importantly, mental health problems associated with the pandemic are not only limited to COVID-19 patients but also affect the general population, including vulnerable groups such as healthcare workers and people with chronic conditions [10]. Moreover, recent observational studies suggest the long-term mental health sequelae of COVID-19, including post-traumatic stress disorder (PTSD), anxiety, and depression [11,12]. Accordingly, in the era of COVID-19, mental health is considered to be a high public health priority [13].

In the context of COVID-19, ICT has become increasingly popular as a coping tool during the pandemic [14]. Advanced ICT services such as telepresence service, remote monitoring service, virtual visit, and environmental disinfection are designed to meet the needs of this pandemic and help individuals and clinicians [15]. In addition, mindfulness-based approaches are considered to be beneficial in mitigating the short- and long-term negative impact of the pandemic on human mental health [16]. This is because mindfulness training cultivates an individual’s ability to cope with the various and widespread stresses experienced in life, and this ability is needed in the COVID-19 or post-COVID-19 era [17]. Additionally, mindfulness has the potential to be involved in addressing collective mental health challenges due to the COVID-19 pandemic, not only at the individual level, but also by engaging in social processes with collective mindfulness [18]. Encouragingly, mindfulness combined with ICT has played a role, especially in improving mental health, during the COVID-19 pandemic [19]. Moreover, some countries, such as South Korea, have been offering mind–body modalities, including mindfulness, through telemedicine services to the general population and individuals with COVID-19 during the pandemic [20].

Bibliometric analysis is now firmly established as a scientific specialty, and new trends in topics of interest, competing groups, and possibilities for cooperation can be identified through the analysis [21]. In other words, bibliometric analysis explores research topics of interest, uncovers and provides an overview of research trends, and promotes related research [21]. Google Trends can be used to analyze health trends and measure public interest in topics of interest [22]. Analysis of this database is also being utilized in other academic areas, not limited to health, including social science [23,24]. This approach can be used to better understand public health behavior through big data analysis; thus, mindfulness during the COVID-19 pandemic may be an appropriate topic for analysis. In addition, in terms of big data utilization, Google Trends analysis has the potential to be used not only for simple monitoring but also for forecasting [25,26]. Mindfulness, either by itself or in combination with ICT, is gaining popularity in the context of COVID-19, and this is not limited to the medical field. Therefore, investigating both research and public interest in mindfulness in the era of COVID-19 may help establish health policies and research directions in this field.

The purpose of this study was to investigate the public and research interest in mindfulness in the context of COVID-19 by Google Trends and bibliometric analysis, respectively. The COVID-19 pandemic has put pressure on the public to regard health as a global public good [27], and public interest and dissemination of information in the public have become an important basis for health policy in the era of COVID-19 [28]. Although the bibliometric analysis technique provides information limited to a pictorial view of the relevant fields, associated keywords, prominent authors, institutions, citing patterns, and global cooperation among authors, it is considered valuable as a tool to support future research directions and, thereby, future R&D decision-making [29]. Therefore, investigating the public interest, as well as the research interest in mindfulness through this study, will provide a perspective on the value and research direction of mindfulness from the perspective of public mental health in the future. In particular, filling the gap between public interest and research interest will provide an important research direction for establishing policies to manage public mental health in the COVID-19 and post-COVID-19 era.

## 2. Materials and Methods

### 2.1. Google Trends Analysis

#### 2.1.1. Data Sources

Google Trends is a service that aggregates search terms used on Google’s search engine, the results of which are anonymized and categorized. When comparing two or more search terms in this service, the relative search popularity can be compared within a range of 0–100 points (i.e., relative search volume [RSV]). An RSV of 100 means the highest number of searches during a given period, and an RSV of less than 100 is calculated as the proportion of the highest number of searches. For example, an RSV of 50 means half the number of searches compared to an RSV of 100.

#### 2.1.2. Analytical Strategy

Since Google Trends provides data from 2004 and this analysis was conducted in December 2022, the search period was December 2004 to November 2022, a total of 18 years. In addition to analyzing the entire period, the period was subdivided into 6 periods of 3 years each (i.e., Period 1: December 2004–November 2007; Period 2: December 2007–November 2010; Period 3: December 2010–November 2013; Period 4: December 2013–November 2016; Period 5: December 2016–November 2019; Period 6: December 2019–November 2022). The reason why the period was subdivided into three years is because Period 6 was a three-year period related to COVID-19. Specifically, as Period 6 is 3 years from the time severe acute respiratory syndrome coronavirus 2 (SARS-CoV-2) was discovered, this analysis was able to examine changes in public interest in mindfulness before and after the COVID-19 era. Thus, the 3-year subdivision was expected to be useful for examining public interest in mindfulness related to COVID-19.

To further investigate the longitudinal variation in public interest in mindfulness, four related topics were searched, including ‘Mental health’, ‘Psychotherapy’, ‘Psychoactive substance’, and ‘Antidepressants’. In the case of ‘Mindfulness’, it was searched as a search term because it was not classified as a topic in Google Trends. The RSV trend lines of ‘Mindfulness’ and the four related topics were calculated for the entire period, and the degree of change was quantified based on the slope value. In addition, Pearson’s correlation coefficient was calculated to analyze the correlation between the RSVs of the search results. The correlation coefficient was tested for statistical power with a 2-tailed test, and *p* < 0.05 was considered statistically significant. Finally, ‘Top related topics’ and ‘Top related queries’ for the search term ‘Mindfulness’ were investigated for each period (1st to 6th period), and the popularity of this search term according to the region was also investigated. The data collected from Google Trends were downloaded from the web page as csv format files, and statistical analysis and visualization were performed using Microsoft Excel 365 (Microsoft Corporation, Redmond, WA, USA) and SPSS version 18 (SPSS Inc., Chicago, IL, USA). Specifically, in the calculation of Pearson’s correlation coefficient, RSVs of the search results were regarded as variables in the software SPSS. The correlation coefficient was calculated to identify the linear correlation between these variables, and the type was selected as Pearson.

### 2.2. Bibliometric Analysis

#### 2.2.1. Data Sources

The search database for bibliometric analysis was the Science Citation Index Expanded database and Social Sciences Citation Index database of the Web of Science Core Collection. The search terms used were determined by referring to previously published bibliometric studies in this field [30,31]. Specifically, COVID-19 and related terms were searched in the title search field, and mindfulness was searched in the topic (i.e., title, abstract, or keywords) search field. In addition, the literature type was limited to “Article”. The search queries were as follows: (TI = (COVID-19) OR TI = (Coronavirus disease 2019) OR TI = (COVID-2019) OR TI = (2019-nCoV) OR TI = (nCov-2019) OR TI = (SARS-CoV) OR TI = (Severe acute respiratory syndrome coronavirus 2) OR TI = (Novel Coronavirus)) AND (TS = (mindfulness*) OR TS = (mindful*)) AND DT = (Article). The search date was 10 January 2023, and there was no limitation on time span and language. The bibliographic information retrieved from the web page was downloaded as a “plain text” file in txt format with the option of “full record with cited references”. No duplications were found among the documents retrieved. A total of 435 articles were included. The bibliographic data were extracted as full records with cited references.

#### 2.2.2. Analytical Strategy

For bibliometric analysis, VOSviewer version 1.6.18 (Centre for Science and Technology Studies, Leiden University, Leiden, The Netherlands) [32], a software that analyzes bibliometric data and visualizes networks, was used. This freely available software allows for easier understanding and analysis of network data by providing a visual representation of the complex linking structure of bibliographies. Additionally, the software helps identify key articles, authors, and institutions related to specific keywords and topics. The plain text file downloaded from the Web of Science Core Collection were opened and analyzed in this software. To understand research trends in this field, the most frequently cited articles were selected from the retrieved articles. The year of publication, study design, related clinical topics, target population, journal of publication, and total number of citations of the selected articles were analyzed. In addition, the number of publications in this field according to the country was analyzed.

To investigate important research topics in this field and their changes over time, keyword co-occurrence analysis and overlay visualization were conducted. For the keyword co-occurrence analysis, a two-dimensional keyword map was constructed using VOSviewer software. The unit of analysis was ‘all keywords’, the counting method was ‘full counting’, and the minimum number of occurrences of a keyword was ‘five’. In this map, the size of the node reflects the frequency of the keyword, and the weight of the connection line indicates the number of articles in which keywords co-occur. Clustering was performed at a resolution of 1.00 with a minimum cluster size of 40, and individual clusters were differentiated by colors including red, blue, yellow, and green. The two-dimensional keyword map was converted and presented using the overlay visualization function, through which changes in research on this topic according to the timeline of COVID-19 were visualized. Moreover, cooperation pattern analysis analyzed the current state of collaboration in mindfulness research in the context of COVID-19 among authors, institutions, and countries. Clustering was performed at a resolution of 1.00 with a minimum cluster size of 10. The connection strength between nodes was calculated as the total link strength (TLS). TLS indicates how closely a node is connected to other nodes, and the higher the value, the stronger the link.

## 3. Results

### 3.1. Trends of Public Interest in Mindfulness

Based on the R^2^ values, the most suitable trend line models were adopted, and their slopes were examined for the five RSV trajectories (Appendix A). Only the trend line for ‘Mindfulness’ was adopted as linear given that its search popularity has steadily increased. However, the search popularity of the other four topics slightly decreased over time before increasing again. Overall, ‘Mental health’ had the highest RSV in all sub-periods. The RSV for this term showed a gradually decreasing trend since 2004, which was the lowest from 2010 to 2013; subsequently, it was greatly increased until November 2022. The RSVs of the remaining four topics showed a slight and gradual increase until November 2022. Visually, public interest in ‘Mindfulness’ and the other four topics in the context of COVID-19 showed the sharpest increase in ‘Mental health’ (Figure 1).

Correlation analysis of the five RSV trajectories was conducted. Throughout the entire period, the RSVs of four topics except for ‘Mindfulness’ showed a statistically significant positive correlation with each other (r = 0.344 to 0.834; all *p* < 0.001). This trend was also consistently observed in the sub-periods (Appendix A). However, the correlation between the RSV trajectory of ‘Mindfulness’ and that of other topics was different. Specifically, throughout the entire period (i.e., December 2004 to November 2022), the RSV trajectory of ‘Mindfulness’ showed a statistically significant negative correlation with that of ‘Psychotherapy’ (r = −0.432, *p* < 0.001). In addition, in the 6th period (i.e., December 2019 to November 2022), the period after the discovery of SARS-CoV-2, the RSV trajectory of ‘Mindfulness’ showed a statistically significant negative correlation with that of ‘Antidepressants’ (r = −0.470, *p* = 0.004). In other words, a significant negative correlation between public interests in ‘Mindfulness’ and that of ‘Antidepressants’, which had not been previously discovered, was found in the COVID-19 era (Table 1).

Throughout the entire period, ‘meditation’ and ‘mindfulness meditation’ were the queries most related to the search term ‘Mindfulness’. There were no new related queries found in the 6th period. However, notably, ‘youtube’ appeared as a related query since the 4th period. Furthermore, throughout the entire period, ‘Meditation’ was the topic most related to the search term ‘Mindfulness’. ‘Psychological stress’ was a related topic until the 3rd period; however, ‘Anxiety’ was one of the top five related topics in the 6th period. In other words, since the middle of 2010, public interest in mindfulness contents via YouTube has increased, and in the era of COVID-19, anxiety, a clinical symptom along with mindfulness, has begun to receive attention as related topics to mindfulness (Table 2).

### 3.2. Trends of Research Interest in Mindfulness

#### 3.2.1. Frequently Cited Articles

The top 10 most frequently cited articles [33,34,35,36,37,38,39,40,41,42] were examined. Three studies (3/10, 30%) [35,37,42] directly mentioned mindfulness in the title. Six of the studies (6/10, 60%) [33,35,37,39,40,41] were cross-sectional studies. One of these studies (1/10, 10%) [42] was an intervention study, a before–after study in which mindfulness meditation was provided as an intervention for female teachers. The clinical topic of one study [34] was considered not directly related to mindfulness because it was about operational directives. Of the remaining nine studies, there were four studies (4/9, 44.44%) [33,35,39,40] on psychological stress (including psychological flexibility), one study (1/9, 11.11%) [36] on occupational burnout and PTSD, one study (1/9, 11.11%) [37] on job insecurity and emotional exhaustion, and one study (1/9, 11.11%) [38] on wellness. The remaining two studies (2/9, 22.22%) [41,42] were about three or more mental health-related aspects. The target population of five studies (55.56%) [36,37,38,41,42] was limited to specific occupational groups. Among these studies, healthcare workers including clinicians were common in two studies (2/9, 22.22%) [36,38], and restaurant workers [37], college students [41], and teachers [42] were the target population in one study each (1/9, 11.11%). Of the 435 studies retrieved, 308 studies were cited once or more (a total of 4390 citations). Among them, the total number of citations of the top 10 most frequently cited articles was 1377 (31.37%) (Table 3).

#### 3.2.2. Publications of Articles

Among the 435 articles retrieved, 339 articles (77.93%) were published in five countries. According to the number of publications, the top five countries were the United States (n = 149, 34.25%), China (n = 74, 17.01%), UK (n = 51, 11.72%), Italy (n = 34, 7.82%), and Spain (n = 31, 7.13%). In particular, articles published in the United States and China accounted for more than 50% of the total retrieved articles (Figure 2a). Among the 435 articles retrieved, 123 articles (28.28%) were published in five journals. According to the number of publications, the top five journals were International Journal of Environmental Research and Public Health (n = 48, 11.03%), Frontiers in Psychology (n = 38, 8.74%), Frontiers in Psychiatry (n = 16, 3.68%), Mindfulness (n = 11, 2.53%), and Frontiers in Public Health and PLoS ONE (n = 10, 2.30%, each) (Figure 2b).

#### 3.2.3. Keyword Co-Occurrence Analysis

A total of 182 keywords were found in five or more articles, and the top five most common keywords were ‘COVID-19’ (n = 255, 58.62%), ‘mindfulness’ (n = 226, 51.95%), ‘depression’ (n = 113, 25.98%), ‘anxiety’ (n = 113, 25.98%), and ‘stress’ (n = 111, 25.52%). The keyword co-occurrence network was divided into four clusters. Cluster 1 (red) was related to ‘mindfulness’ and showed connectivity across the searched keywords. The keywords unique to Cluster 1 included ‘burnout’, ‘nurses’, ‘satisfaction’, ‘emotional regulation’, ‘life’, and ‘benefits’. Cluster 2 (blue) was related to ‘COVID-19’ and showed connectivity across the searched keywords. The keywords unique to Cluster 2 included ‘pandemic’, ‘coronavirus’, ‘telehealth’, ‘health’, and ‘care’. ‘Stress’ belonged to Cluster 2; however, it was a large node and had a topology close to Cluster 1. Cluster 3 (yellow) was related to ‘mental health’ and showed a topology closer to Cluster 1. On the other hand, Cluster 4 (green) was related to ‘anxiety and depression’ and showed a topology closer to Cluster 2. The keywords unique to Cluster 3 included ‘youth’, ‘self-compassion’, and ‘scale’. The keywords unique to Cluster 4 included ‘disorder’, ‘psychological distress’, ‘therapy’, and ‘psychometric properties’ (Figure 3a). In the keyword map visualized with the overlay visualization function, there was no noticeable difference between each keyword according to the timeline of COVID-19, but ‘mental health’ seemed to be a relatively recent research trend, compared to other important nodes such as ‘mindfulness’, ‘COVID-19’, and ‘stress’ (Figure 3b).

#### 3.2.4. Cooperation Pattern Analysis

The cooperation pattern of authors on mindfulness research was analyzed. The cooperation pattern network was based the authors of at least two documents on mindfulness. A total of 141 authors were included in the network, but the clusters were not connected to each other. The size of the largest set of connected authors was 13. In other words, it suggests that cooperation among authors on mindfulness research in the context of COVID-19 is insufficient. The top five authors with the strongest link strength were all Spanish researchers, including Miquel Bennasar-Veny (University of the Balearic Islands, Spain) (TLS: 25), Mauro Garcia Toro (University of the Balearic Islands, Spain) (TLS: 25), Pablo Alonso coello (Iberoamerican Cochrane Centre, Spain) (TLS: 23), Javier García Campayo (University of Zaragoza, Spain) (TLS: 23), and Elena Gervilla (University of the Balearic Islands, Spain) (TLS: 23) (Figure 4a).

The cooperation pattern of organizations on this topic was also sporadic. A total of 178 organizations of at least two documents were included in the network, and the size of the largest set of connected organizations was 95. The top five institutions with the strongest link strength were mostly United States organizations, including Johns Hopkins University (United States) (TLS: 13), Rutgers University (United States) (TLS: 13), University of Manitoba (Canada) (TLS: 13), University of Nottingham (England) (TLS: 13), and Harvard Medical School (United States) (TLS: 12) (Figure 4b).

Finally, a total of 45 countries of at least two documents were included in the network, and all nodes were connected. The top five countries with the strongest link strength were United States (TLS: 101), England (TLS: 93), China (TLS: 56), Canada (TLS: 48), and Italy (TLS: 46). Three clusters were found in this network. The largest Cluster 1 (red) has the core node of the United States, the second largest Cluster 2 (green) has the core node of Spain, and the smallest Cluster 3 (blue) has the core node of France (Figure 4c).

## 4. Discussion

This study aimed to examine the trends of public and research interest in mindfulness in the context of COVID-19. Through Google Trends analysis, public interest in mindfulness was monitored and quantified, and through bibliographic analysis, research interest in mindfulness was analyzed and visualized.

### 4.1. Trends of Public Interest in Mindfulness

Google Trend analysis revealed that the overall RSV of ‘Mental health’, a topic potentially related to mindfulness, was increased over time. In contrast, the RSVs of ‘Mindfulness’, ‘Psychotherapy’, ‘Psychoactive substance’, and ‘Antidepressants’ were only increased slightly. The study period (December 2004 to November 2022) was subdivided into 6 periods of 3 years, and the 6th period was 3 years from the discovery of SARS-CoV-2 in December 2019. This study aimed to investigate the change in public interest in mindfulness related to COVID-19 by focusing on the difference between indicators analyzed in the 6th period and the previous periods.

Notable correlations between the RSV of the search term ‘Mindfulness’ and that of the topics ‘Psychotherapy’ and ‘Antidepressants’ were observed. Specifically, the RSVs of ‘Mindfulness’ and ‘Psychotherapy’ showed an overall significant negative correlation (r = −0.432) but a statistically significant positive correlation in the 4th (r = 0.541) and 5th (r = 0.739) periods. This finding may be explained by the popularization of mindfulness-based psychotherapy, such as MBCT; however, there is no evidence to confirm this possibility. Nevertheless, it is interesting that the search volume of MBCT in MEDLINE on PubMed (Appendix A) and the RSV of MBCT in Google Trends (Appendix A) were increased during this period.

The RSVs of ‘Mindfulness’ and ‘Antidepressants’ showed an overall significant positive correlation (r = 0.485) but a statistically significant negative correlation in the 6th period (r = −0.470). As the current study does not suggest causality for the associations found and does not rule out contingency, only some assumptions can be made about these associations. Mindfulness-based interventions (MBIs) have been shown to help improve several psychiatric disorders, including depressive disorders [43]. Accordingly, MBIs have been considered as a promising adjuvant option for depression treatment in clinical settings [44]. However, a recent meta-analysis of 30 randomized controlled trials found that a standardized MBI (MBCT) could have beneficial effects similar to those of cognitive behavioral therapy in the treatment of depression [45]. In addition, MBIs combined with ICT have been reported to be effective in improving mental health, including depression, anxiety, and stress, in the context of the COVID-19 pandemic [46]. These findings suggest that MBIs may be regarded as an alternative rather than an adjuvant to antidepressants in the treatment of depression. In addition to the accumulated clinical evidence on MBIs, media and publicity may have contributed to the public perception of mindfulness [47]. The difficulty in accessing mental health providers due to the COVID-19 pandemic may also explain the findings [48].

Based on the analysis results of the top related queries and topics for the search term ‘Mindfulness’, there were no new related queries in the COVID-19 era (i.e., 6th period). However, although psychological stress was popular as a topic related to mindfulness before the COVID-19 era (i.e., 1st to 5th period), anxiety appeared to be a popular related topic in the COVID-19 era. A meta-analysis of 43 community-based studies found that the rates of anxiety in the general population could be more than three times higher during the COVID-19 pandemic [49], making anxiety a major mental health problem associated with COVID-19. Based on the top related queries and topics of the current study, mindfulness may have initially gained popularity as a management method for psychological stress. However, lately, especially in the era of COVID-19, it is possible that mindfulness is gaining popularity for managing psychiatric symptoms such as anxiety.

### 4.2. Trends of Research Interest in Mindfulness

The 10 most popular articles [33,34,35,36,37,38,39,40,41,42] were identified among the 435 articles retrieved, and the majority of studies (60%) [33,35,37,39,40,41] were cross-sectional studies. On the other hand, only one intervention study [42] was included as the most popular articles. Excluding one study [34] not directly related to mindfulness, the majority of studies (55.56%) [33,35,36,39,40] were on stress-related conditions, including psychological stress and PTSD. In addition, the majority of them (55.56%) [36,37,38,41,42] targeted specific occupational groups, such as healthcare workers, restaurant workers, college students, and teachers. These findings can be interpreted as indicating that intervention studies of MBIs have not yet had a significant impact on this research field of mindfulness in the era of COVID-19 and may further suggest a potential gap with public interest in mindfulness (e.g., managing psychiatric symptoms such as anxiety).

Articles in this field were published the most in the United States and China (51.26%). According to the results of the cooperation pattern analysis, the United States and China were establishing the largest cluster of the cooperation pattern of countries on mindfulness research topics. However, the cooperation between authors and institutions in this field was not international, but sporadically. The largest network of cooperation among authors was centered in Spain, and that of cooperation between institutions was centered on institutions located in the United States. However, many countries share a common mental health burden from COVID-19 [50], and cooperation to address it is encouraged. Consistently, a recent bibliometric analysis of mindfulness research pointed to geographic inequalities in research in this field and highlighted the need for more collaboration [51]. Although mindfulness is considered to have originated in oriental religious practices, MBIs have demonstrated their mental health benefits in a variety of populations in different countries, so it is worth researching mindfulness as a resource to improve human mental health in the COVID-19 or post-COVID-19 period [17]. For example, a panel of psychiatrists from 15 countries, through the Delphi consensus, proposed a protocol for telemental health care during this pandemic, and MBI was considered its first-line intervention [52].

According to keyword co-occurrence analysis, the most common keywords in the included studies were ‘COVID-19’, ‘mindfulness’, ‘depression’, ‘anxiety’, and ‘stress’. Clustering identified four clusters in the constructed keyword co-occurrence network, and each cluster was formed around ‘mindfulness’, ‘COVID-19’, ‘mental health’, and ‘anxiety and depression’. ‘Mindfulness’ and ‘COVID-19’ accounted for the largest and most widespread nodes, showing relevance in almost all domains. The ‘Stress’ node, classified as a cluster of COVID-19, was of significant size, and it was considered to be closely related to mindfulness based on its topology. On the other hand, the ‘mental health’ and ‘anxiety and depression’ clusters were smaller than those of ‘mindfulness’ and ‘COVID-19’. The ‘mental health’ cluster had a topology closer to that of ‘mindfulness’, and the ‘anxiety and depression’ cluster had a topology closer to that of ‘COVID-19’. Analysis of the related nodes demonstrated that the ‘mental health’ cluster included the non-clinical population and some psychological elements that may be associated with mindfulness, such as self-compassion. The ‘anxiety and depression’ cluster tended to include clinical elements such as disorder, psychological distress, and therapy.

Articles on mindfulness in the context of COVID-19 were closely related to depression, anxiety, stress, and mental health. The findings are consistent with the positive association between daily SARS-CoV-2 infection rates and the major depressive disorder and anxiety disorder prevalence observed during the COVID-19 pandemic [50]. In addition, according to the clusters, studies with more clinical relevance may focus on anxiety or depression in the context of COVID-19, and studies on self-care for non-clinical or sub-clinical populations may focus on other aspects of mental health in the context of mindfulness.

### 4.3. Implications on the Public and Research Fields

This study analyzed both public and research interest in mindfulness in the context of COVID-19. The results suggest a growing public interest in managing psychiatric symptoms during COVID-19, with mindfulness as an intervention and possibly an alternative to antidepressants, along with a growing public interest in mental health. However, most of the studies that had the most important impact in this field were cross-sectional studies, and intervention studies using MBI were lacking. In addition, despite the emphasis on global efforts to address the mental health burden in the COVID-19 era [50], research in this field showed a lack of multi-national cooperation. Encouragingly, however, our keyword co-occurrence analysis found anxiety and depression as important clinical elements related to mindfulness. The implications of this study highlight resolving the gap between the public interest and research interests found, and further interventional, ideally multi-national, studies of MBIs on anxiety or depression in the context of COVID-19 may be encouraged.

Our Google Trends analysis also found an increased public interest in mindfulness contents via YouTube since the middle of 2010. Some studies found a critical link between mental health and YouTube content during the COVID-19 pandemic [53,54]. Nevertheless, according to our bibliographic analysis, studies in this field do not seem to consider online platforms such as YouTube as an important research topic in the context of mindfulness research. Therefore, to fill this gap, an attempt to investigate the use of MBIs via online platforms including YouTube in the public and demonstrate their effectiveness and safety on mental health of individuals could be encouraged.

### 4.4. Strengths and Limitations

This study has the strength of suggesting current views and future research directions in this field by combining and analyzing bibliometrics and Google Trends data. In particular, the proposed future research directions are based on bridging the gap between public and research interest in mindfulness in the context of COVID-19. Given that public interest and dissemination of information to the public are an important basis for health policy [28], and that strategies to maintain the integrity of the mental health of individuals should be implemented at the social level in the era of COVID-19 [50], this study has public health relevance. Perhaps the methodology attempted in this study can be used as a reference for researchers to investigate public interest and research interest in a specific topic in the future. Even in that case, the interpretation of the results should focus on discovering the gap between the two interest types (i.e., public and research interest), discussing the importance of filling the gap, and providing a solution strategy to fill the gap.

However, some limitations should be acknowledged. First, due to the nature of Google Trends and bibliometric analysis, the causality of the findings cannot be explored, and the possibility of discovery due to chance cannot be ruled out. In addition, as Google Trends can be influenced by a country’s Internet penetration and the relative popularity of Internet search engines, it may be premature to interpret the findings from this study as indicative of a global public interest in mindfulness. Second, as the search term ‘Mindfulness’ was searched in English in Google Trends, the findings may only be meaningful in countries where the term is searched in English. In addition, there was no topic for ‘Mindfulness’ in the Google Trends system, limiting comparability with other topics. Third, as the search database for bibliometric analysis in this study was limited to the Web of Science, it is possible that some related articles were missed.

## 5. Conclusions

This study conducted Google Trends and bibliometric analysis to investigate public and research interest in mindfulness related to COVID-19. As a result, the RSV for ‘Mindfulness’ showed a slight increase over the entire period. In particular, from December 2019 to November 2022, the three years belonging to the COVID-19 period, the RSV trajectory of ‘Mindfulness’ showed a statistically significant negative correlation with that of ‘Antidepressants’. The majority of most popular articles in this field were cross-sectional studies. It was found that most studies in this field were published in the United States and China. However, the cooperation between authors and institutions in this field was sporadically. In a network map, four clusters were identified (i.e., ‘mindfulness’, ‘COVID-19’, ‘anxiety and depression’, and ‘mental health’) in the studies. However, one of the important limitations of this study is that our methodology does not prove the potential causality of the findings.

According to the findings and the limitations, further interventional, ideally multi-national, studies of MBIs on anxiety or depression in the context of COVID-19 may be encouraged. Additionally, research investigating the public use of MBIs via online platforms such as YouTube, and verifying its effectiveness and safety, may also be encouraged. The findings may provide insights into potential areas of interest and identify ongoing trends in this field. Additionally, the findings of this study may be referenced by policy makers responsible for future R&D decision-making on the mental health burden of COVID-19.

## Figures and Tables

**Figure 1 ijerph-20-03807-f001:**
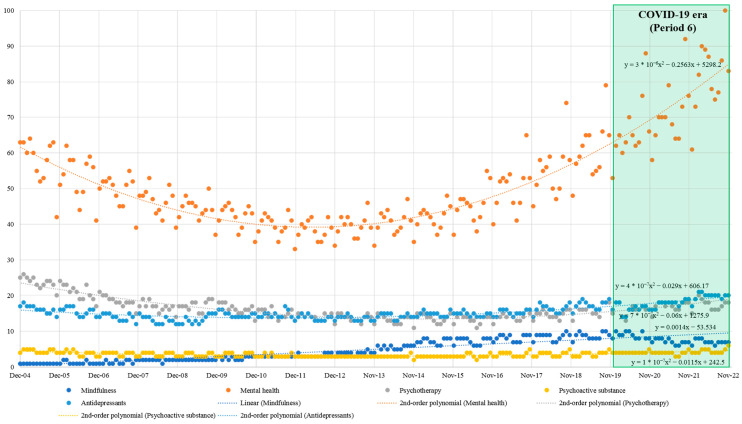
Relative search volume of ‘Mindfulness’, ‘Mental health’, ‘Psychotherapy’, ‘Psychoactive substance’, and ‘Antidepressants’ (2004 to 2022, monthly) in Google Trends. Note. The y-axis represents the relative search score and is distributed between 0 and 100 points. The x-axis represents the period.

**Figure 2 ijerph-20-03807-f002:**
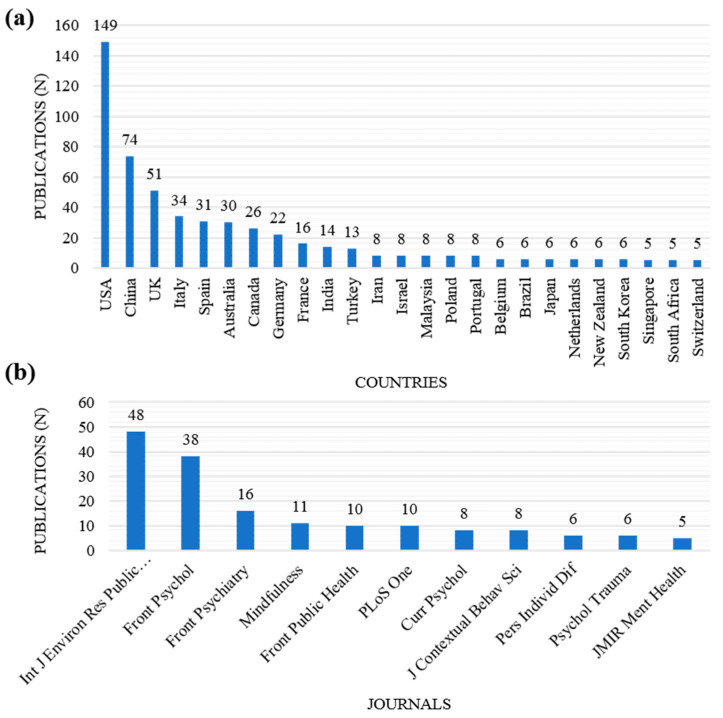
(**a**) Countries and (**b**) Journals with more than five articles published on mindfulness during the study period.

**Figure 3 ijerph-20-03807-f003:**
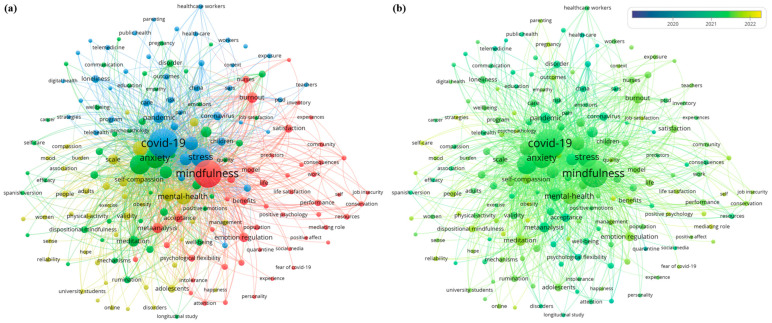
(**a**) Keyword co-occurrence network graph and (**b**) Overlay visualization network graph of mindfulness research topics in the COVID-19 era. Note. Red cluster: mindfulness; Blue cluster: COVID-19; Yellow cluster: mental health; Green cluster: anxiety and depression.

**Figure 4 ijerph-20-03807-f004:**
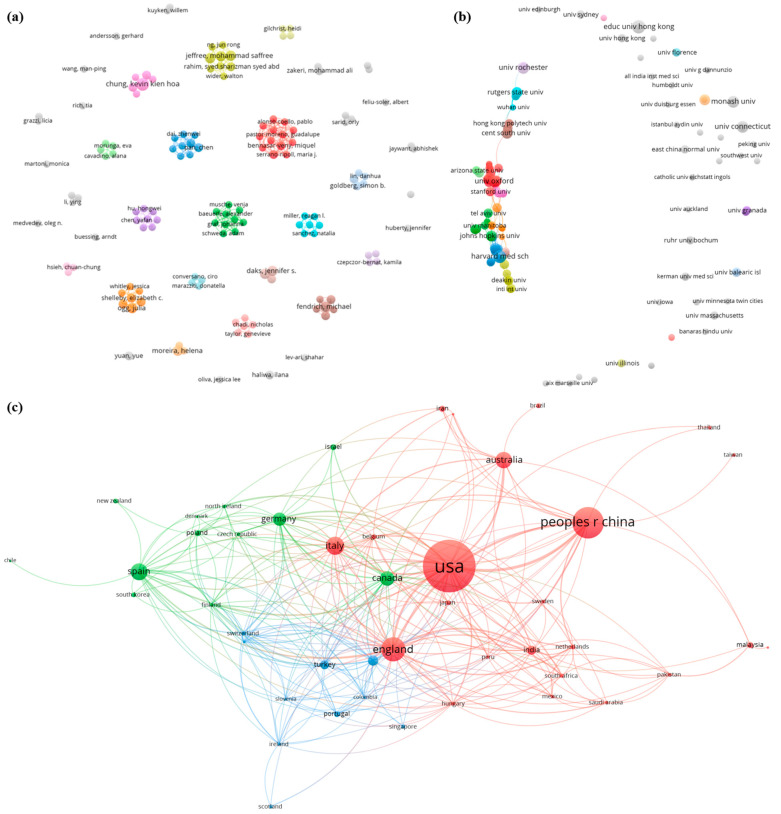
The cooperation pattern of (**a**) authors, (**b**) organizations, and (**c**) countries on mindfulness research topics in the COVID-19 era.

**Table 1 ijerph-20-03807-t001:** Pearson’s correlation between the search term ‘Mindfulness’ and four other search terms over time.

Correlation		Mental Health	Psychotherapy	Psychoactive Substance	Antidepressants
Mindfulness	Total	0.412 **	−0.432 **	0.055	0.485 **
	P1	0.011	−0.142	0.069	−0.115
	P2	−0.030	0.120	0.332 *	0.430 **
	P3	0.306	−0.033	−0.230	−0.214
	P4	0.695 **	0.541 **	0.223	0.527 **
	P5	0.781 **	0.739 **	0.559 **	0.688 **
	P6	−0.235	−0.056	−0.015	−0.470 **

**Note**. * indicates *p* < 0.05; ** indicates *p* < 0.01 (two-tailed test). P1 = Period 1: December 2004–November 2007; P2 = Period 2: December 2007–November 2010; P3 = Period 3: December 2010–November 2013; P4 = Period 4: December 2013–November 2016; P5 = Period 5: December 2016–November 2019; P6 = Period 6: December 2019–November 2022.

**Table 2 ijerph-20-03807-t002:** Top queries and topics related to the search term ‘Mindfulness’ over time.

**Related Queries**	**P1**	**P2**	**P3**	**P4**	**P5**	**P6**
Top 1	meditation	meditation	meditation	meditation mindfulness	mindfulness meditation	mindfulness meditation
Top 2	mindfulness meditation	mindfulness meditation	mindfulness meditation	meditation	meditation	meditation
Top 3	mindfulness therapy	mindfulness therapy	stress	mindfulness training	mindfulness youtube	what is mindfulness
Top 4	mindfulness stress reduction	mindfulness stress	training mindfulness	stress	kids mindfulness	kids mindfulness
Top 5	mindfulness	training mindfulness	mindfulness therapy	youtube mindfulness	what is mindfulness	mindfulness meaning
**Related Topics**	**P1**	**P2**	**P3**	**P4**	**P5**	**P6**
Top 1	Meditation	Meditation	Meditation	Meditation	Meditation	Meditation
Top 2	Therapy	Therapy	Therapy	Exercise	Sati	Sati
Top 3	Mindfulness-based stress reduction	Training	Training	Training	Course	Exercise
Top 4	Psychological stress	Mindfulness-based stress reduction	Mindfulness-based stress reduction	Course	Exercise	Therapy
Top 5	Training	Psychological stress	Psychological stress	Therapy	Training	Anxiety

Note. P1 = Period 1: December 2004–November 2007; P2 = Period 2: December 2007–November 2010; P3 = Period 3: December 2010–November 2013; P4 = Period 4: December 2013–November 2016; P5 = Period 5: December 2016–November 2019; P6 = Period 6: December 2019–November 2022.

**Table 3 ijerph-20-03807-t003:** Top 10 most frequently cited articles.

Title	Publication Year	Study Design	Relevant Clinical Topics (Target Population)	Journal of Publication	Total Citation
Stress and parenting during the global COVID-19 pandemic	2020	Cross-sectional study	Psychological stress (parents with a child under the age of 18 years)	Child Abuse and Neglect	485
Surgery in COVID-19 patients: operational directives	2020	Commentary	Others: operational directives (not specific)	World Journal of Emergency Surgery	183
Mindfulness, Age and Gender as Protective Factors Against Psychological Distress During COVID-19 Pandemic	2020	Cross-sectional study	Psychological stress (general population)	Frontiers in Psychology	125
Occupational burnout syndrome and post-traumatic stress among healthcare professionals during the novel coronavirus disease 2019 (COVID-19) pandemic	2020	Commentary	Occupational burnout syndrome and PTSD (healthcare professionals)	Best Practice and Research Clinical Anesthesiology	114
Do mindfulness and perceived organizational support work? Fear of COVID-19 on restaurant frontline employees’ job insecurity and emotional exhaustion	2021	Literature review and cross-sectional study	Job insecurity and emotional exhaustion (restaurant frontline employees)	International Journal of Hospitality Management	99
Clinician Wellness During the COVID-19 Pandemic: Extraordinary Times and Unusual Challenges for the Allergist/Immunologist	2020	Commentary	Wellness (clinicians)	The Journal of Allergy and Clinical Immunology: In Practice	90
The moderating roles of psychological flexibility and inflexibility on the mental health impacts of COVID-19 pandemic and lockdown in Italy	2020	Cross-sectional study	Psychological flexibility and inflexibility (general population)	Journal of Contextual Behavioral Science	77
Psychological flexibility and inflexibility as sources of resiliency and risk during a pandemic: Modeling the cascade of COVID-19 stress on family systems with a contextual behavioral science lens	2020	Cross-sectional study	Psychological flexibility and inflexibility (general population)	Journal of Contextual Behavioral Science	72
Psychiatric symptoms, risk, and protective factors among university students in quarantine during the COVID-19 pandemic in China	2021	Cross-sectional study	Psychiatric symptoms including anxiety, depression, and traumatic stress (university students)	Globalization and Health	68
Positive Impact of Mindfulness Meditation on Mental Health of Female Teachers during the COVID-19 Outbreak in Italy	2020	Before-after study	Mindfulness skills, empathy, personality profiles, interoceptive awareness, psychological well-being, emotional distress, and burnout levels (female teachers)	International Journal of Environmental Research and Public Health	64

Abbreviations. COVID-19, Coronavirus disease 2019; PTSD, post-traumatic stress disorder.

## Data Availability

The data used to support the findings of this study are included within the article.

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
