# Peer review of "Research and Public Interest in Mindfulness in the COVID-19 and Post-COVID-19 Era: A Bibliometric and Google Trends Analysis"

_ijerph, 2023, doi:10.3390/ijerph20053807_

Round 1

Reviewer 1 Report

The introductory section offers a robust theoretical foundation for the study, establishing enough evidence of the essence of conducting google trends and bibliometric analysis on mindfulness as a single entity or in relation to ICT especially during the Covid-19 period. It scholarly exposes the research lacuna in comprehending public and research interests on mindfulness and how the novel findings will inform health policies, amongst others. 

The methods section is well discussed and scholarly, comprehensive enough to allow replication. However, the author(s) must explain with enough justification the selection of the timelines (December 2004-November 2022). 

The results section comprehensively presents the results of the google trend and bibliometric analysis conducted with illustrations using tables and figures.

The concluding section needs to be improved. After introducing the study's purpose, state the key findings and the tentative conclusions drawn from them. After this, sum up the suggested possible areas of further research.

Author Response

  • Response to Comments from Reviewer 1

Comment 1:

The introductory section offers a robust theoretical foundation for the study, establishing enough evidence of the essence of conducting google trends and bibliometric analysis on mindfulness as a single entity or in relation to ICT especially during the Covid-19 period. It scholarly exposes the research lacuna in comprehending public and research interests on mindfulness and how the novel findings will inform health policies, amongst others.

Response 1:        

Thank you for your careful review and insightful comments that have significantly enhanced our manuscript.

Comment 2:

The methods section is well discussed and scholarly, comprehensive enough to allow replication. However, the author(s) must explain with enough justification the selection of the timelines (December 2004-November 2022).

Response 2:        

Thank you for your comment. We further described why we set the total timeline (December 2004 to November 2022) and why we subdivided it by 3 years, in this revised manuscript.

“Since Google Trends provides its data from 2004 and this analysis was conducted in December 2022, the search period was December 2004 to November 2022, a total of 18 years. In addition to analyzing the entire period, the period was subdivided into 6 periods of 3 years each (i.e., Period 1: 2004.12. – 2007.11.; Period 2: 2007.12. – 2010.11.; Period 3: 2010.12. – 2013.11.; Period 4: 2013.12. – 2016.11.; Period 5: 2016.12. – 2019.11.; Period 6: 2019.12. – 2022.11.). The reason why the period was subdivided into three years is because Period 6 was a three-year period related to COVID-19. Specifically, as Period 6 is 3 years from the time severe acute respiratory syndrome coronavirus 2 (SARS-CoV-2) was discovered, this analysis was able to examine changes in public interest in mindfulness before and after the COVID-19 era. Thus, the 3-year subdivision was expected to be useful for examining public interest in mindfulness related to COVID-19.”

(Please refer to the red text on page 3.)

Comment 3:

The results section comprehensively presents the results of the google trend and bibliometric analysis conducted with illustrations using tables and figures.

Response 3:        

Thank you for your comment.

Comment 4:

The concluding section needs to be improved. After introducing the study's purpose, state the key findings and the tentative conclusions drawn from them. After this, sum up the suggested possible areas of further research.

Response 4:        

Thank you for your comment. As the reviewer commented, the Conclusion section was further improved. Specifically, after the purpose of this study, the tentative conclusions drawn were described. Then, areas available for further study were summarized and described.

“This study conducted Google Trends and bibliometric analysis to investigate public and research interest in mindfulness related to COVID-19. As a result, the RSV for ‘Mindfulness’ showed a slight increase over the entire period. In particular, from December 2019 to November 2022, the three years belonging to the COVID-19 period, the RSV trajectory of ‘Mindfulness’ showed a statistically significant negative correlation with that of ‘Antidepressants’. The majority of most popular articles in this field were cross-sectional studies. It was found that most studies in this field were published in the United States and China. However, the cooperation between authors and institutions in this field was sporadically. In a network map, four clusters were identified (i.e., ‘mindfulness’, ‘COVID-19’, ‘anxiety and depression’, and ‘mental health’) in the studies. However, one of the important limitations of this study is that our methodology does not prove the potential causality of the findings.

According to the findings and the limitations, further interventional, ideally multi-national, studies of MBIs on anxiety or depression in the context of COVID-19 may be encouraged. Also, researches investigating the public use of MBIs via online platforms such as YouTube and verifying its effectiveness and safety may also be encouraged. The findings may provide insights into potential areas of interest and identify ongoing trends in this field. Also, the findings of this study may be referenced by policy makers responsible for future R&D decision-making on the mental health burden of COVID-19.”

(Please refer to the red text on pages 13-14.)

Reviewer 2 Report

Title: (Research and Public Interest in Mindfulness in the COVID-19 2 Era: A Bibliometric and Google Trends Analysis)

Comment 1: The author is advised to revise the research title, as it seems not contemporary. The covid phenomenon has now become a thing of the past. A title that reflects the research inclination of mindfulness in the post covid world would be suitable.

Abstract:

In lines 8-9, is the interest of the "public" necessary to mention?

In lines 14-15, the author should clarify how the two-dimensional keyword map was constructed.

In lines 38-40, please clarify the "pooled prevalence of DASS" for whom, any specific population, gender, etc.?

In the introduction, the author discusses/highlights the background of individual mindfulness, a concept developed by (Sternberg, 2000; Ryle, 1949; Langer, 1989), but somehow the connection between mindfulness, DASS elements, COVID-19, and ICT is not clear. The author is advised to include the following in the introduction:

a.        Why is individual mindfulness chosen for this research and relevant to the post-covid era?

b.       How this research would help facilitate the propagation of mindfulness research about the DASS (depression, anxiety, stress, etc.).

c.        Since the bibliometric analysis technique only provides a pictorial view of the relevant fields, associated keywords, prominent authors, institutions, citing patterns, and global cooperations among authors, how come this brief research will aid/beef up the current research on individual mindfulness?

d.       The author is advised to add two to three bullet points to highlight such research intentions/objectives.

The author is also advised to add 2-3 paragraphs of literature on individual mindfulness that attributes its utility to coping with stress, etc., DASS. It will help increase readers' interest in individual mindfulness and its possible benefits. The author can take a brief overview of individual mindfulness from this seminal work (Sutcliffe, K. M., Vogus, T. J., & Dane, E. (2016). Mindfulness in Organizations: A Cross-Level Review. Annual Review of Organizational Psychology and Organizational Behavior, 3(1), Article 1. https://doi.org/10.1146/annurev-orgpsych-041015-062531)

Analysis 2.1.2: I am not sure why the research period was divided into six periods, as, in general, from the web of science, we can download the report for the total desired period; the author should clarify this a bit further.

From Lines 92-94: How Pearson's correlation coefficient was calculated to analyze the correlation between the RSVs of the search results? Through which software, and how. The author should elaborate in detail as to what method was opted to conduct this nature of data to such analysis.

2.2.1. Data sources: the author should clarify and specifically mention the term they used to extract the data from the web of science for the analysis and the exclusion and inclusion criteria for the literature/articles.

2.2.1 Figure 1. It needs to be concluded as to what happened to mindfulness research and what change has occurred due to covid! The author should include the critical area where change is visible/noticeable due to covid over mindfulness research.

2.2.1. Table 1 and Table 2 are presented fine, but again what this table means, instead of using technical language, an author should summarise the key findings in a brief paragraph.

The authors have provided only two analyses, i.e., 3.2.1. Frequently cited articles, 3.2.2. Keyword co-occurrence analysis, but to enhance the depth and width of the article, the author is advised to add the following analysis to their article:

The Cooperation Pattern of authors on mindfulness research, leading publishing bodies on mindfulness research, Global cooperation network on mindfulness research in the covid or post-covid era, Co-citation analysis of mindfulness research, and most importantly, the term analysis with the timeline for mindfulness research.

By adding the analysis mentioned above, the author can increase the contribution to the literature. All such analyses can be performed from the already downloaded database from WOS. The author may also seek guidance from the following works for referencing:

a.        Saleem, M. S., Isha, A. S. N., Yusop, Y. M., Awan, M. I., & Naji, G. M. A. (2023). Mindfulness Research: A Bibliometric Analysis. In B. Alareeni & A. Hamdan (Eds.), Innovation of Businesses, and Digitalization during Covid-19 Pandemic (Vol. 488, pp. 611–632). Springer International Publishing. https://doi.org/10.1007/978-3-031-08090-6_38

b.       van Nunen, K., Li, J., Reniers, G., & Ponnet, K. (2018). Bibliometric analysis of safety culture research. Safety Science, 108, 248–258. https://doi.org/10.1016/j.ssci.2017.08.011

Discussion and analysis. In general, the discussion is okay that is describing the statistical part, but the author should highlight the actual contribution that is being made through this research as follows:

a.        What are the implications of most cited articles for the research community?

b.       What are the indications and signs for research and the general community regarding the keyword co-occurrence analysis?

c.        Are there any possibilities for future research areas where based on this analysis, funding can be secured?

d.       How should individuals who have been through covid/post-covid world view this research?

Implications: The author is advised to insert 1-2 paragraphs to highlight their research findings and provide some practical contribution of this research towards the research community and the individuals in general.

The conclusions are fine but need to be revised based on the corrections given above.

Author Response

  • Response to Comments from Reviewer 2

Comment 1:

The author is advised to revise the research title, as it seems not contemporary. The covid phenomenon has now become a thing of the past. A title that reflects the research inclination of mindfulness in the post covid world would be suitable.

Response 1:        

Thank you for your careful review. Like the reviewer's comment, it might be more appropriate to name it ‘post COVID world’ for current researches. However, our study also includes acute breakout periods of COVID-19, such as late 2019 and early 2020. Therefore, in this revised manuscript, the title was modified as follows to include the meaning of the acute breakout and post COVID periods.

“Research and public interest in mindfulness in the COVID-19 and post-COVID-19 era: A bibliometric and Google trends”

(Please refer to the red text on page 1.)

Comment 2:

Abstract:

1) In lines 8-9, is the interest of the "public" necessary to mention?

2) In lines 14-15, the author should clarify how the two-dimensional keyword map was constructed.

3) In lines 38-40, please clarify the "pooled prevalence of DASS" for whom, any specific population, gender, etc.?

Response 2:        

Thank you for your comment.

1) Yes, it is necessary to be mentioned. That's because this study investigated both the ‘public’ interest as well as the ‘research’ interest in mindfulness. This was achieved with separate methods: bibliometric analysis and Google Trends analysis, respectively.

2) As the maximum limit of abstracts of this journal is 200 words, there was insufficient description in the abstract content. As the reviewer commented, a two-dimensional keyword map construction method was added to this revised manuscript.

“Keyword co-occurrence analysis was conducted, and a two-dimensional keyword map was constructed using VOSviewer software.”

(Please refer to the red text on page 1.)

3) In this revised manuscript, we describe the ‘whom’ of pooled prevalence of DASS.

“According to a comprehensive systematic review of the prevalence of mental health problems during the COVID-19 pandemic, the overall pooled prevalence of depression, anxiety, distress, and insomnia reached 31%, 32%, 41%, and 38%, respectively, among all types of target population (e.g., general population, medical staffs, students, patients with COVID-19, and patients with other clinical conditions) [8].”

(Please refer to the red text on pages 1-2.)

Comment 3:

In the introduction, the author discusses/highlights the background of individual mindfulness, a concept developed by (Sternberg, 2000; Ryle, 1949; Langer, 1989), but somehow the connection between mindfulness, DASS elements, COVID-19, and ICT is not clear. The author is advised to include the following in the introduction:

  1. Why is individual mindfulness chosen for this research and relevant to the post-covid era?
  2. How this research would help facilitate the propagation of mindfulness research about the DASS (depression, anxiety, stress, etc.).
  3. Since the bibliometric analysis technique only provides a pictorial view of the relevant fields, associated keywords, prominent authors, institutions, citing patterns, and global cooperations among authors, how come this brief research will aid/beef up the current research on individual mindfulness?
  4. The author is advised to add two to three bullet points to highlight such research intentions/objectives.

Response 3:        

Thank you for your kind comment. The reviewer kindly explained at each point what the Introduction of this manuscript should be revised. Following the reviewer's recommendation, we have amended the Introduction in this revised manuscript as follows.

“Structured mindfulness training programs such as mindfulness-based stress reduction and mindfulness-based cognitive therapy (MBCT) have contributed to the popularization of mindfulness [5]. The popularity of mindfulness in modern society is related to increased stress and anxiety in modern society, as well as a growing public interest in well-being [6]. The popularity of mindfulness is extending not only to individuals, but also to businesses and organizations (e.g., corporate mindfulness) [7,8].

In the context of COVID-19, ICT has become increasingly popular as a coping tool during the pandemic [14]. Advanced ICT services such as telepresence service, remote monitoring service, virtual visit, and environmental disinfection are designed to meet the needs of this pandemic and help individuals and clinicians [15]. In addition, mindfulness-based approaches are considered to be beneficial in mitigating the short- and long-term negative impact of the pandemic on human mental health [16]. This is because mindfulness training cultivates an individual's ability to cope with the various and widespread stresses experienced in life, and this ability is needed in the COVID-19 or post-COVID-19 era [17]. Also, mindfulness has the potential to be involved in addressing collective mental health challenges due to the COVID-19 pandemic, not only at the individual level, but also by engaging in social processes with collective mindfulness [18]. Encouragingly, mindfulness combined with ICT has played a role, especially in improving mental health, during the COVID-19 pandemic [19]. Moreover, some countries, such as South Korea, have been offering mind-body modalities, including mindfulness, through telemedicine services to the general population and individuals with COVID-19 during the pandemic [20].

The purpose of this study was to investigate the public and research interest in mindfulness in the context of COVID-19 by Google Trends and bibliometric analysis, respectively. The COVID-19 pandemic has put pressure on the public to regard health as a global public good [27], and public interest and dissemination of information in the public have become the important basis for health policy in the era of COVID-19 [28]. Although the bibliometric analysis technique provides information limited to a pictorial view of the relevant fields, associated keywords, prominent authors, institutions, citing patterns, and global cooperation among authors, it is considered valuable as a tool to support future research directions and, thereby, future R&D decision-making [29]. Therefore, investigating the public interest as well as the research interest in mindfulness through this study will provide a perspective on the value and research direction of mindfulness from the perspective of public mental health in the future. In particular, filling the gap between public interest and research interest will provide an important research direction for establishing policies to manage public mental health in the COVID-19 and post-COVID-19 era.”

(Please refer to the red text on pages 1-3.)

Comment 4:

The author is also advised to add 2-3 paragraphs of literature on individual mindfulness that attributes its utility to coping with stress, etc., DASS. It will help increase readers' interest in individual mindfulness and its possible benefits. The author can take a brief overview of individual mindfulness from this seminal work (Sutcliffe, K. M., Vogus, T. J., & Dane, E. (2016). Mindfulness in Organizations: A Cross-Level Review. Annual Review of Organizational Psychology and Organizational Behavior, 3(1), Article 1. https://doi.org/10.1146/annurev-orgpsych-041015-062531)

Response 4:        

Thank you for your comment. The document recommended by the reviewer is cited in this revised manuscript with the following sentence:

“The popularity of mindfulness in modern society is related to increased stress and anxiety in modern society, as well as a growing public interest in well-being [6]. The popularity of mindfulness is extending not only to individuals, but also to businesses and organizations (e.g., corporate mindfulness) [7,8].

This is because mindfulness training cultivates an individual's ability to cope with the various and widespread stresses experienced in life, and this ability is needed in the COVID-19 or post-COVID-19 era [17]. Also, mindfulness has the potential to be involved in addressing collective mental health challenges due to the COVID-19 pandemic, not only at the individual level, but also by engaging in social processes with collective mindfulness [18].”

[18] Sutcliffe, K.M.; Vogus, T.J.; Dane, E. Mindfulness in organizations: A cross-level review. Annual Review of Organizational Psychology and Organizational Behavior 2016, 3, 55-81.

(Please refer to the red text on pages 1-2.)

Comment 5:

Analysis 2.1.2: I am not sure why the research period was divided into six periods, as, in general, from the web of science, we can download the report for the total desired period; the author should clarify this a bit further.

Response 5:        

Thank you for your comment. We further described why we set the total timeline (December 2004 to November 2022) and why we subdivided it by 3 years, in this revised manuscript.

“Since Google Trends provides its data from 2004 and this analysis was conducted in December 2022, the search period was December 2004 to November 2022, a total of 18 years. In addition to analyzing the entire period, the period was subdivided into 6 periods of 3 years each (i.e., Period 1: 2004.12. – 2007.11.; Period 2: 2007.12. – 2010.11.; Period 3: 2010.12. – 2013.11.; Period 4: 2013.12. – 2016.11.; Period 5: 2016.12. – 2019.11.; Period 6: 2019.12. – 2022.11.). The reason why the period was subdivided into three years is because Period 6 was a three-year period related to COVID-19. Specifically, as Period 6 is 3 years from the time severe acute respiratory syndrome coronavirus 2 (SARS-CoV-2) was discovered, this analysis was able to examine changes in public interest in mindfulness before and after the COVID-19 era. Thus, the 3-year subdivision was expected to be useful for examining public interest in mindfulness related to COVID-19.”

(Please refer to the red text on page 3.)

Comment 6:

From Lines 92-94: How Pearson's correlation coefficient was calculated to analyze the correlation between the RSVs of the search results? Through which software, and how. The author should elaborate in detail as to what method was opted to conduct this nature of data to such analysis.

Response 6:        

Thank you for your comment. As the reviewer commented, the calculation method of Pearson's correlation coefficient and the software used are described in this revised manuscript.

“The data collected from Google Trends were downloaded from the web page as csv format files, and statistical analysis and visualization were performed using Microsoft Excel 365 (Microsoft Corporation, Redmond, WA, USA) and SPSS version 18 (SPSS Inc., Chicago, IL, USA). Specifically, in the calculation of Pearson's correlation coefficient, RSVs of the search results were regarded as variables in the software SPSS. The correlation coefficient was calculated to identify the linear correlation between these variables, and the type was selected as Pearson.”

(Please refer to the red text on page 3.)

Comment 7:

2.2.1. Data sources: the author should clarify and specifically mention the term they used to extract the data from the web of science for the analysis and the exclusion and inclusion criteria for the literature/articles.

Response 7:        

Thank you for your comment. We have added a description of the bibliographic information obtained from the web of science as follows.

“The search date was January 10, 2023, and there was no limitation on time span and language. The bibliographic information retrieved from the web page was downloaded as a “plain text” file in txt format with the option of “full record with cited references”. No duplications were found among the documents retrieved. A total of 435 articles were included.

The plain text file downloaded from the Web of Science Core Collection were opened and analyzed in this software.”

(Please refer to the red text on page 4.)

Comment 8:

2.2.1 Figure 1. It needs to be concluded as to what happened to mindfulness research and what change has occurred due to covid! The author should include the critical area where change is visible/noticeable due to covid over mindfulness research.

Response 8:        

Thank you for your comment. We modified Figure 1 in this revised manuscript. Specifically, the COVID-19 era window has been added in green shade to the existing Figure 1.

(Please refer to the Figure 1 on page 5.)

Comment 9:

2.2.1. Table 1 and Table 2 are presented fine, but again what this table means, instead of using technical language, an author should summarise the key findings in a brief paragraph.

Response 9:        

Thank you for your comment. We added the following sentences to the main findings summarized for Tables 1 and 2.

“… In addition, in the 6th period (i.e., 2019.12. to 2022.11.), the period after the discovery of SARS-CoV-2, the RSV trajectory of ‘Mindfulness’ showed a statistically significant negative correlation with that of ‘Antidepressants’ (r = -0.470, p = 0.004). In other words, a significant negative correlation between public interests in ‘Mindfulness’ and that of ‘Antidepressants’, which had not been previously discovered, was found in the COVID-19 era (Table 1).”

(Please refer to the red text on page 5.)

“… ‘Psychological stress’ was a related topic until the 3rd period; however, ‘Anxiety’ was one of the top five related topics in the 6th period. In other words, since the middle of 2010, public interest in mindfulness contents via YouTube has increased, and in the era of COVID-19, anxiety, a clinical symptom along with mindfulness, has begun to receive attention as related topics to mindfulness (Table 2).”

(Please refer to the red text on page 6.)

Comment 10:

The authors have provided only two analyses, i.e., 3.2.1. Frequently cited articles, 3.2.2. Keyword co-occurrence analysis, but to enhance the depth and width of the article, the author is advised to add the following analysis to their article:

The Cooperation Pattern of authors on mindfulness research, leading publishing bodies on mindfulness research, Global cooperation network on mindfulness research in the covid or post-covid era, Co-citation analysis of mindfulness research, and most importantly, the term analysis with the timeline for mindfulness research.

By adding the analysis mentioned above, the author can increase the contribution to the literature. All such analyses can be performed from the already downloaded database from WOS. The author may also seek guidance from the following works for referencing:

  1. Saleem, M. S., Isha, A. S. N., Yusop, Y. M., Awan, M. I., & Naji, G. M. A. (2023). Mindfulness Research: A Bibliometric Analysis. In B. Alareeni & A. Hamdan (Eds.), Innovation of Businesses, and Digitalization during Covid-19 Pandemic (Vol. 488, pp. 611–632). Springer International Publishing. https://doi.org/10.1007/978-3-031-08090-6_38
  2. van Nunen, K., Li, J., Reniers, G., & Ponnet, K. (2018). Bibliometric analysis of safety culture research. Safety Science, 108, 248–258. https://doi.org/10.1016/j.ssci.2017.08.011

Response 10:      

Thank you for your comment. In this revised manuscript, as recommended by the reviewer, the following additional bibliographic analysis was performed. Accordingly, figures related to the additional analysis have been added.

1) We analyzed the top journals in which studies in this field were published.

“Among the 435 articles retrieved, 123 articles (28.28%) were published in five journals. According to the number of publications, the top five journals were International Journal of Environmental Research and Public Health (n = 48, 11.03%), Frontiers in Psychology (n = 38, 8.74%), Frontiers in Psychiatry (n = 16, 3.68%), Mindfulness (n = 11, 2.53%), and Frontiers in Public Health and PLoS One (n = 10, 2.30%, each) (Figure 2(b)).”

(Please refer to Figure 2(b) and the red text on page 8.)

2) In addition to the existing keyword co-occurrence analysis, we additionally analyzed changes in keyword popularity according to the COVID-19 timeline through an overlay visualization function.

“In the keyword map visualized with the overlay visualization function, there was no noticeable difference between each keyword according to the timeline of COVID-19, but ‘mental health’ seemed to be a relatively recent research trend, compared to other important nodes such as ‘mindfulness’, ‘COVID-19’, and ‘stress’ (Figure 3(b)).”

(Please refer to Figure 3(b) and the red text on page 9.)

3) Cooperation pattern analysis of authors, organizations, and countries has been added.

“3.2.4. Cooperation pattern analysis

The cooperation pattern of authors on mindfulness research was analyzed. The cooperation pattern network was based the authors of at least two documents on mindfulness. A total of 141 authors were included in the network, but the clusters were not connected to each other. The size of the largest set of connected authors was 13. In other words, it suggests that cooperation among authors on mindfulness research in the context of COVID-19 is insufficient. The top five authors with the strongest link strength were all Spanish researchers, including Miquel Bennasar-Veny (University of the Balearic Islands, Spain) (TLS: 25), Mauro Garcia Toro (University of the Balearic Islands, Spain) (TLS: 25), Pablo Alonso coello (Iberoamerican Cochrane Centre, Spain) (TLS: 23), Javier García Campayo (University of Zaragoza, Spain) (TLS: 23), and Elena Gervilla (University of the Balearic Islands, Spain) (TLS: 23) (Figure 4(a)).

The cooperation pattern of organizations on this topic was also sporadic. A total of 178 organizations of at least two documents were included in the network, and the size of the largest set of connected organizations was 95. The top five institutions with the strongest link strength were mostly United States organizations, including Johns Hopkins University (United States) (TLS: 13), Rutgers University (United States) (TLS: 13), University of Manitoba (Canada) (TLS: 13), University of Nottingham (England) (TLS: 13), and Harvard Medical School (United States) (TLS: 12) (Figure 4(b)).

Finally, a total of 45 countries of at least two documents were included in the network, and all nodes were connected. The top five countries with the strongest link strength were United States (TLS: 101), England (TLS: 93), China (TLS: 56), Canada (TLS: 48), and Italy (TLS: 46). Three clusters were found in this network. The largest Cluster 1 (red) has the core node of the United States, the second largest Cluster 2 (green) has the core node of Spain, and the smallest Cluster 3 (blue) has the core node of France (Figure 4(c)).”

(Please refer to Figure 4 and the red text on pages 9-10.)

Comment 11:

Discussion and analysis. In general, the discussion is okay that is describing the statistical part, but the author should highlight the actual contribution that is being made through this research as follows:

  1. What are the implications of most cited articles for the research community?
  2. What are the indications and signs for research and the general community regarding the keyword co-occurrence analysis?
  3. Are there any possibilities for future research areas where based on this analysis, funding can be secured?
  4. How should individuals who have been through covid/post-covid world view this research?

Response 11:      

Thank you for your comment. Each of the editor's kind comments were reflected in this revised manuscript as follows.

  1. The 10 most popular articles [33-42] were identified among the 435 articles retrieved, and the majority of studies (60%) [33,35,37,39-41] were cross-sectional studies. On the other hand, only one intervention study [42] was included as the most popular articles. Excluding one study [34] not directly related to mindfulness, the majority of studies (55.56%) [33,35,36,39,40] were on stress-related conditions including psychological stress and PTSD. In addition, the majority of them (55.56%) [36-38,41,42] targeted specific occupational groups, such as healthcare workers, restaurant workers, college students, and teachers. These findings can be interpreted as indicating that intervention studies of MBIs have not yet had a significant impact on this research field of mindfulness in the era of COVID-19, and may further suggest a potential gap with public interest in mindfulness (e.g., managing psychiatric symptoms such as anxiety).

(Please refer to the red text on page 12.)

  1. “Encouragingly, however, our keyword co-occurrence analysis found anxiety and depression as important clinical elements related to mindfulness. The implications of this study highlight resolving the gap between the public interest and research interests found, and further interventional, ideally multi-national, studies of MBIs on anxiety or depression in the context of COVID-19 may be encouraged.”

(Please refer to the red text on page 13.)

  1. “This study has the strength of suggesting current view as well as future research directions in this field by combining and analyzing bibliometrics and Google Trends data. In particular, the proposed future research directions are based on bridging the gap between public and research interest in mindfulness in the context of COVID-19. Given that public interest and dissemination of information to the public are an important basis for health policy [28], and that strategies to maintain the integrity of the mental health of individuals should be implemented at the social level in the era of COVID-19 [50], this study has public health relevance.”

(Please refer to the red text on page 13.)

  1. 4.3. Implications on the public and research fields

This study analyzed both public and research interest in mindfulness in the context of COVID-19. The results suggest a growing public interest in managing psychiatric symptoms during COVID-19, with mindfulness as an intervention and possibly an alternative to antidepressants, along with a growing public interest in mental health. However, most of the studies that had the most important impact in this field were cross-sectional studies, and intervention studies using MBI were lacking. In addition, despite the emphasis on global efforts to address the mental health burden in the COVID-19 era [50], researches in this field showed a lack of multi-national cooperation. Encouragingly, however, our keyword co-occurrence analysis found anxiety and depression as important clinical elements related to mindfulness. The implications of this study highlight resolving the gap between the public interest and research interests found, and further interventional, ideally multi-national, studies of MBIs on anxiety or depression in the context of COVID-19 may be encouraged.

Our Google Trends analysis also found the increased public interest in mindfulness contents via YouTube since the middle of 2010. And some studies found the critical link between mental health and YouTube content during the COVID-19 pandemic [52,53]. Nevertheless, according to our bibliographic analysis, studies in this field do not seem to consider online platforms such as YouTube as an important research topic, in the context of mindfulness research. Therefore, to fill this gap, an attempt to investigate the use of MBIs via online platforms including YouTube in the public and demonstrate their effectiveness and safety on mental health of individuals could be encouraged.”

(Please refer to the red text on pages 12-13.)

Comment 12:

Implications: The author is advised to insert 1-2 paragraphs to highlight their research findings and provide some practical contribution of this research towards the research community and the individuals in general.

Response 12:      

Thank you for your comment. Implications on the public and research fields section has been added as follows:

4.3. Implications on the public and research fields

This study analyzed both public and research interest in mindfulness in the context of COVID-19. The results suggest a growing public interest in managing psychiatric symptoms during COVID-19, with mindfulness as an intervention and possibly an alternative to antidepressants, along with a growing public interest in mental health. However, most of the studies that had the most important impact in this field were cross-sectional studies, and intervention studies using MBI were lacking. In addition, despite the emphasis on global efforts to address the mental health burden in the COVID-19 era [50], researches in this field showed a lack of multi-national cooperation. Encouragingly, however, our keyword co-occurrence analysis found anxiety and depression as important clinical elements related to mindfulness. The implications of this study highlight resolving the gap between the public interest and research interests found, and further interventional, ideally multi-national, studies of MBIs on anxiety or depression in the context of COVID-19 may be encouraged.

Our Google Trends analysis also found the increased public interest in mindfulness contents via YouTube since the middle of 2010. And some studies found the critical link between mental health and YouTube content during the COVID-19 pandemic [52,53]. Nevertheless, according to our bibliographic analysis, studies in this field do not seem to consider online platforms such as YouTube as an important research topic, in the context of mindfulness research. Therefore, to fill this gap, an attempt to investigate the use of MBIs via online platforms including YouTube in the public and demonstrate their effectiveness and safety on mental health of individuals could be encouraged.”

(Please refer to the red text on pages 12-13.)

Comment 13:

The conclusions are fine but need to be revised based on the corrections given above.

Response 13:      

Thank you for your comment. Based on the above amendments, the Conclusions have been amended as follows.

“This study conducted Google Trends and bibliometric analysis to investigate public and research interest in mindfulness related to COVID-19. As a result, the RSV for ‘Mindfulness’ showed a slight increase over the entire period. In particular, from December 2019 to November 2022, the three years belonging to the COVID-19 period, the RSV trajectory of ‘Mindfulness’ showed a statistically significant negative correlation with that of ‘Antidepressants’. The majority of most popular articles in this field were cross-sectional studies. It was found that most studies in this field were published in the United States and China. However, the cooperation between authors and institutions in this field was sporadically. In a network map, four clusters were identified (i.e., ‘mindfulness’, ‘COVID-19’, ‘anxiety and depression’, and ‘mental health’) in the studies. However, one of the important limitations of this study is that our methodology does not prove the potential causality of the findings.

According to the findings and the limitations, further interventional, ideally multi-national, studies of MBIs on anxiety or depression in the context of COVID-19 may be encouraged. Also, researches investigating the public use of MBIs via online platforms such as YouTube and verifying its effectiveness and safety may also be encouraged. The findings may provide insights into potential areas of interest and identify ongoing trends in this field. Also, the findings of this study may be referenced by policy makers responsible for future R&D decision-making on the mental health burden of COVID-19.”

(Please refer to the red text on pages 13-14.)

Reviewer 3 Report

Review of “Research and Public Interest in Mindfulness in the COVID-19 Era: A Bibliometric and Google Trends Analysis”

Thanks for providing me with the opportunity to read your manuscript,  which deals with a very relevant and timely topic in the Covid-19 era.

While I believe the manuscript has potential, I believe it needs some further development. Below follows some comments and suggestions that I hope could be useful in a revision of the paper:

 ·         The attempt to combine bibliometrics and Google Trends data is interesting, and you should underline this as one of the contributions of your paper.

·         In general, you should articulate the contribution(s) of the study more clearly in the introduction (and reiterate it in the conclusion section). What is new and value-adding in your study? Maybe separate between theoretical, methodological, and practical contributions. 

·         I think you could provide some more background to the concept of Mindfulness and its current popularity in society. This would be useful for non-specialized readers. It is probably perceived by some as quite hard to grasp and quite fuzzy.

·         You could mention that mindfulness is popular in businesses and organizations too, e.g. “corporate mindfulness” [1,2]. Maybe you could also add some reflections on mindfulness from a more critical perspective.

·         I think you could also provide some more background to Google Trends and cite some more research on the applications of this tool in different academic areas (not only health), see for example [3-6].

·         Explain more clearly why chose to carry out a keyword co-occurrence analysis (and not one of the many other types of bibliometric analyses).

·         I think you should provide a citation for the VOSviewer software [7]. Also, why was this software package chosen and not one of the alternatives (e.g. Biblioshiny, Citespace).

·         The conclusion section (5) is too short and short be expanded. See other comments.

·         Maybe a possibility would be to move the discussion of limitations to the conclusion section. I would also link the discussion of limitations to a discussion of future research directions.

·         In my view, ideas for future work would be particularly interesting and useful. In particular, it would be great if you could come up with some more ideas for combining bibliometrics and Google Trends analysis in the future (as well as possible issues researchers may face).

·         Not sure if the contribution section is needed since there seems to be only one author. However, this is up to the editors.

Good luck!

References

1.            Wrenn, M.V. From Mad to Mindful: Corporate Control Through Corporate Spirituality. Journal of Economic Issues 2020, 54, 503-509, doi:10.1080/00213624.2020.1756660.

 2.            Khanna, V.; Khanna, P.D. Critical perspectives on corporate mindfulness and workplace spirituality. In Spirituality in Management; Springer: 2019; pp. 179-193.

 3.            Silva, E.S.; Hassani, H.; Madsen, D.Ø.; Gee, L. Googling Fashion: Forecasting Fashion Consumer Behaviour Using Google Trends. Social Sciences 2019, 8, 111.

4.            Dinis, G.; Breda, Z.; Costa, C.; Pacheco, O. Google Trends in tourism and hospitality research: a systematic literature review. Journal of Hospitality and Tourism Technology 2019.

5.            Jun, S.-P.; Yoo, H.S.; Choi, S. Ten years of research change using Google Trends: From the perspective of big data utilizations and applications. Technological Forecasting and Social Change 2018, 130, 69-87.

6.            Choi, H.; Varian, H. Predicting the present with google trends. Economic Record 2012, 88, 2-9.

7.            Van Eck, N.J.; Waltman, L. Software survey: VOSviewer, a computer program for bibliometric mapping. Scientometrics 2010, 84, 523-538.

Author Response

  • Response to Comments from Reviewer 3

Comment 1:

Thanks for providing me with the opportunity to read your manuscript,  which deals with a very relevant and timely topic in the Covid-19 era.

While I believe the manuscript has potential, I believe it needs some further development. Below follows some comments and suggestions that I hope could be useful in a revision of the paper:

Response 1:        

Thank you for your careful review and insightful comments that have significantly enhanced our manuscript.

Comment 2:

  • The attempt to combine bibliometrics and Google Trends data is interesting, and you should underline this as one of the contributions of your paper.
  • In general, you should articulate the contribution(s) of the study more clearly in the introduction (and reiterate it in the conclusion section). What is new and value-adding in your study? Maybe separate between theoretical, methodological, and practical contributions.

Response 2:        

Thank you for your comment. We have further emphasized the analyzes attempted in this study in the revised manuscript.

“The purpose of this study was to investigate the public and research interest in mindfulness in the context of COVID-19 by Google Trends and bibliometric analysis, respectively. The COVID-19 pandemic has put pressure on the public to regard health as a global public good [27], and public interest and dissemination of information in the public have become the important basis for health policy in the era of COVID-19 [28]. Although the bibliometric analysis technique provides information limited to a pictorial view of the relevant fields, associated keywords, prominent authors, institutions, citing patterns, and global cooperation among authors, it is considered valuable as a tool to support future research directions and, thereby, future R&D decision-making [29]. Therefore, investigating the public interest as well as the research interest in mindfulness through this study will provide a perspective on the value and research direction of mindfulness from the perspective of public mental health in the future. In particular, filling the gap between public interest and research interest will provide an important research direction for establishing policies to manage public mental health in the COVID-19 and post-COVID-19 era.”

(Please refer to the red text on pages 2-3.)

“This study has the strength of suggesting current view as well as future research directions in this field by combining and analyzing bibliometrics and Google Trends data. In particular, the proposed future research directions are based on bridging the gap between public and research interest in mindfulness in the context of COVID-19. Given that public interest and dissemination of information to the public are an important basis for health policy [28], and that strategies to maintain the integrity of the mental health of individuals should be implemented at the social level in the era of COVID-19 [50], this study has public health relevance. Perhaps, the methodology attempted in this study can be used as a reference for researchers to investigate public interest and research interest in a specific topic in the future. Even in that case, the interpretation of the results should focus on discovering the gap between the two interest types (i.e., public and research interest), discussing the importance of filling the gap, and providing a solution strategy to fill the gap.”

(Please refer to the red text on page 13.)

Comment 3:

  • I think you could provide some more background to the concept of Mindfulness and its current popularity in society. This would be useful for non-specialized readers. It is probably perceived by some as quite hard to grasp and quite fuzzy.
  • You could mention that mindfulness is popular in businesses and organizations too, e.g. “corporate mindfulness” [1,2]. Maybe you could also add some reflections on mindfulness from a more critical perspective.

Response 3:        

Thank you for your comment. Based on the reviewer's comments, we added the following description of corporate mindfulness.

“Structured mindfulness training programs such as mindfulness-based stress reduction and mindfulness-based cognitive therapy (MBCT) have contributed to the popularization of mindfulness [5]. The popularity of mindfulness in modern society is related to increased stress and anxiety in modern society, as well as a growing public interest in well-being [6]. The popularity of mindfulness is extending not only to individuals, but also to businesses and organizations (e.g., corporate mindfulness) [7,8].

In the context of COVID-19, ICT has become increasingly popular as a coping tool during the pandemic [14]. Advanced ICT services such as telepresence service, remote monitoring service, virtual visit, and environmental disinfection are designed to meet the needs of this pandemic and help individuals and clinicians [15]. In addition, mindfulness-based approaches are considered to be beneficial in mitigating the short- and long-term negative impact of the pandemic on human mental health [16]. This is because mindfulness training cultivates an individual's ability to cope with the various and widespread stresses experienced in life, and this ability is needed in the COVID-19 or post-COVID-19 era [17]. Also, mindfulness has the potential to be involved in addressing collective mental health challenges due to the COVID-19 pandemic, not only at the individual level, but also by engaging in social processes with collective mindfulness [18]. Encouragingly, mindfulness combined with ICT has played a role, especially in improving mental health, during the COVID-19 pandemic [19]. Moreover, some countries, such as South Korea, have been offering mind-body modalities, including mindfulness, through telemedicine services to the general population and individuals with COVID-19 during the pandemic [20].”

(Please refer to the red text on pages 1-3.)

Comment 4:

  • I think you could also provide some more background to Google Trends and cite some more research on the applications of this tool in different academic areas (not only health), see for example [3-6].

Response 4:        

Thank you for your comment. The documents recommended by the reviewer were cited and the following sentences were added to this revised manuscript.

“Google Trends can be used to analyze health trends and measure public interest in topics of interest [22]. Analysis of this database is also being utilized in other academic areas, not limited to health, including social science [23,24]. This approach can be used to better understand public health behavior through big data analysis; thus, mindfulness during the COVID-19 pandemic may be an appropriate topic for analysis. In addition, in terms of big data utilization, Google Trends analysis has the potential to be used not only for simple monitoring but also for forecasting [25,26].”

(Please refer to the red text on page 2.)

Comment 5:

  • Explain more clearly why chose to carry out a keyword co-occurrence analysis (and not one of the many other types of bibliometric analyses).

Response 5:        

Thank you for your comment. In fact, in this revised manuscript, additional analyzes were conducted in addition to the keyword co-occurrence analysis. We explained the reasons for conducting keyword co-occurrence analysis, along with explanations of additional analyses.

“Additionally, the software helps identify key articles, authors, and institutions related to specific keywords and topics. The plain text file downloaded from the Web of Science Core Collection were opened and analyzed in this software.

To investigate important research topics in this field and their changes over time, keyword co-occurrence analysis and overlay visualization were conducted. For the keyword co-occurrence analysis, a two-dimensional keyword map was constructed using VOSviewer software. The unit of analysis was ‘all keywords’, the counting method was ‘full counting’, and the minimum number of occurrences of a keyword was ‘five’. In this map, the size of the node reflects the frequency of the keyword, and the weight of the connection line indicates the number of articles in which keywords co-occur. Clustering was performed at a resolution of 1.00 with a minimum cluster size of 40, and individual clusters were differentiated by colors including red, blue, yellow, and green. The two-dimensional keyword map was converted and presented using the overlay visualization function, through which changes in research on this topic according to the timeline of COVID-19 were visualized. Moreover, cooperation pattern analysis analyzed the current state of collaboration in mindfulness research in the context of COVID-19 among authors, institutions, and countries. Clustering was performed at a resolution of 1.00 with a minimum cluster size of 10. The connection strength between nodes was calculated as the total link strength (TLS). TLS indicates how closely a node is connected to other nodes, and the higher the value, the stronger the link.”

(Please refer to the red text on page 4.)

Comment 6:

  • I think you should provide a citation for the VOSviewer software [7]. Also, why was this software package chosen and not one of the alternatives (e.g. Biblioshiny, Citespace).

Response 6:        

Thank you for your comment. In this revised manuscript, the citation of VOSviewer software was added, and the reason for using this software was justified.

“For bibliometric analysis, VOSviewer version 1.6.18 (Centre for Science and Technology Studies, Leiden University, Leiden, The Netherlands) [32], a software that analyzes bibliometric data and visualizes networks, was used. This freely available software allows for easier understanding and analysis of network data by providing a visual representation of the complex linking structure of bibliographies.”

(Please refer to the red text on page 4.)

Comment 7:

  • The conclusion section (5) is too short and short be expanded. See other comments.
  • Maybe a possibility would be to move the discussion of limitations to the conclusion section. I would also link the discussion of limitations to a discussion of future research directions.

Response 7:        

Thank you for your comment. In this revised manuscript, the Conclusion section was amended as follows based on the reviewer's comments.

“This study conducted Google Trends and bibliometric analysis to investigate public and research interest in mindfulness related to COVID-19. As a result, the RSV for ‘Mindfulness’ showed a slight increase over the entire period. In particular, from December 2019 to November 2022, the three years belonging to the COVID-19 period, the RSV trajectory of ‘Mindfulness’ showed a statistically significant negative correlation with that of ‘Antidepressants’. The majority of most popular articles in this field were cross-sectional studies. It was found that most studies in this field were published in the United States and China. However, the cooperation between authors and institutions in this field was sporadically. In a network map, four clusters were identified (i.e., ‘mindfulness’, ‘COVID-19’, ‘anxiety and depression’, and ‘mental health’) in the studies. However, one of the important limitations of this study is that our methodology does not prove the potential causality of the findings.

According to the findings and the limitations, further interventional, ideally multi-national, studies of MBIs on anxiety or depression in the context of COVID-19 may be encouraged. Also, researches investigating the public use of MBIs via online platforms such as YouTube and verifying its effectiveness and safety may also be encouraged. The findings may provide insights into potential areas of interest and identify ongoing trends in this field. Also, the findings of this study may be referenced by policy makers responsible for future R&D decision-making on the mental health burden of COVID-19.”

(Please refer to the red text on pages 13-14.)

Comment 8:

  • In my view, ideas for future work would be particularly interesting and useful. In particular, it would be great if you could come up with some more ideas for combining bibliometrics and Google Trends analysis in the future (as well as possible issues researchers may face).

Response 8:        

Thank you for your comment. We described the strengths of the analysis attempted in this study and add some suggestions for future researchers.

“This study has the strength of suggesting current view as well as future research directions in this field by combining and analyzing bibliometrics and Google Trends data. In particular, the proposed future research directions are based on bridging the gap between public and research interest in mindfulness in the context of COVID-19. Given that public interest and dissemination of information to the public are an important basis for health policy [28], and that strategies to maintain the integrity of the mental health of individuals should be implemented at the social level in the era of COVID-19 [50], this study has public health relevance. Perhaps, the methodology attempted in this study can be used as a reference for researchers to investigate public interest and research interest in a specific topic in the future. Even in that case, the interpretation of the results should focus on discovering the gap between the two interest types (i.e., public and research interest), discussing the importance of filling the gap, and providing a solution strategy to fill the gap.”

(Please refer to the red text on page 13.)

Comment 9:

  • Not sure if the contribution section is needed since there seems to be only one author. However, this is up to the editors.

Good luck!

References

  1. Wrenn, M.V. From Mad to Mindful: Corporate Control Through Corporate Spirituality. Journal of Economic Issues 2020, 54, 503-509, doi:10.1080/00213624.2020.1756660.
  2. Khanna, V.; Khanna, P.D. Critical perspectives on corporate mindfulness and workplace spirituality. In Spirituality in Management; Springer: 2019; pp. 179-193.
  3. Silva, E.S.; Hassani, H.; Madsen, D.Ø.; Gee, L. Googling Fashion: Forecasting Fashion Consumer Behaviour Using Google Trends. Social Sciences 2019, 8, 111.
  4. Dinis, G.; Breda, Z.; Costa, C.; Pacheco, O. Google Trends in tourism and hospitality research: a systematic literature review. Journal of Hospitality and Tourism Technology 2019.
  5. Jun, S.-P.; Yoo, H.S.; Choi, S. Ten years of research change using Google Trends: From the perspective of big data utilizations and applications. Technological Forecasting and Social Change 2018, 130, 69-87.
  6. Choi, H.; Varian, H. Predicting the present with google trends. Economic Record 2012, 88, 2-9.
  7. Van Eck, N.J.; Waltman, L. Software survey: VOSviewer, a computer program for bibliometric mapping. Scientometrics 2010, 84, 523-538.

Response 9:        

Thank you for your kind comment.

Round 2

Reviewer 2 Report

The authors have made the required changes and have significantly improved their manuscript.

This manuscript can contribute to the advancement of bibliometric research techniques and can help the propagation of Mindfulness research. However, recent bibliometric analysis references should be cited as follows:

Mindfulness Research: A Bibliometric Analysis. In B. Alareeni & A. Hamdan (Eds.), Innovation of Businesses, and Digitalization during Covid-19 Pandemic (Vol. 488, pp. 611–632). Springer International Publishing. https://doi.org/10.1007/978-3-031-08090-6_38

Author Response

  • Response to Comments from Reviewer 2

Comment 1:

The authors have made the required changes and have significantly improved their manuscript.

This manuscript can contribute to the advancement of bibliometric research techniques and can help the propagation of Mindfulness research. However, recent bibliometric analysis references should be cited as follows:

Mindfulness Research: A Bibliometric Analysis. In B. Alareeni & A. Hamdan (Eds.), Innovation of Businesses, and Digitalization during Covid-19 Pandemic (Vol. 488, pp. 611–632). Springer International Publishing. https://doi.org/10.1007/978-3-031-08090-6_38

Response 1:        

Thank you for your comment. I believe that your comments have greatly improved the quality of this manuscript. And the document recommended by the reviewer is cited in the following appended sentence.

“However, the cooperation between authors and institutions in this field was not international, but sporadically. The largest network of cooperation among authors was centered in Spain, and that of cooperation between institutions was centered on institutions located in the United States. However, many countries share a common mental health burden from COVID-19 [50], and cooperation to address it is encouraged. Consistently, a recent bibliometric analysis of mindfulness research pointed to geographic inequalities in research in this field and highlighted the need for more collaboration [51]. Although mindfulness is considered to have originated in oriental religious practices, MBIs have demonstrated their mental health benefits in a variety of populations in different countries, so it is worth researching mindfulness as a resource to improve human mental health in the COVID-19 or post-COVID-19 period [17].”

[51] Saleem, M.S.; Isha, A.S.N.; Yusop, Y.M.; Awan, M.I.; Naji, G.M.A. Mindfulness Research: A Bibliometric Analysis. In Proceedings of the Innovation of Businesses, and Digitalization during Covid-19 Pandemic: Proceedings of The International Conference on Business and Technology (ICBT 2021), 2022; pp. 611-632.

(Please refer to the red text on page 12.)

Reviewer 3 Report

Thanks for providing a detailed point-by-point response to my comments. The revised version of the manuscript is much improved. Well done. 

Author Response

  • Response to Comments from Reviewer 3

Comment 1:

Thanks for providing a detailed point-by-point response to my comments. The revised version of the manuscript is much improved. Well done.

Response 1:        

Thank you for the comment. I believe that your comments have greatly improved the quality of this manuscript.
